Manuscript prepared for Geosci. Instrum. Method. Data Syst.
with version 2014/09/16 7.15 Copernicus papers of the LaTeX class copernicus.cls.
Date: 7 September 2020

# Easy to build Low power GPS drifters with local storage and GSM modem made from off the shelf components

Rolf Hut[1], Thanda Thatoe Nwe Win[2], and Thom Bogaard[2]

[1]Delft University of Technology, Faculty of Civil Engineering and Geosciences, chair of Water Resources Engineering
[2]Delft University of Technology, Faculty of Civil Engineering and Geosciences, chair of Hydrology

*Correspondence to:* Rolf Hut (r.w.hut@tudelft.nl)

**Abstract.** Drifters that track their position are important tools in studying the hydrodynamic behaviour of rivers. Drifters that can be tracked in real time have so far been rather expensive. Recently due to the rise of the Open Hardware revolution and the associated Arduino ecosystem both GPS receivers and GSM modems have become available at lower prices to 'tinkering scientists', i.e. scientists that like to build their own measurement devices as much as possible. This article serves two goals. Firstly, we provide detailed instructions on how to build a Low Power GPS drifter with local storage and GSM model that we tested in a fieldwork on the confluence of the Chindwin and Ayeyarwady rivers in Myanmar. The device was designed from easily connected off the shelf components, allowing construction without a background in electrical engineering. The instructions allow fellow geoscientists to recreate the device. Secondly, we set the question: "Has the Open Hardware revolution progressed to the point that a low power GPS drifter that wirelessly transmits its position can be made from Open Hardware component by most geoscientists?" We feel this question is relevant and timely as more low-cost Open Hardware devices are promoted but in practice applicability often is restricted to the 'tinkering engineer'. We argue that because of the plug and play nature of the components geoscientist should be able to construct these type of devices. However, to get such devices to operate at low power levels that fieldwork often requires detailed (micro)electrical expertise.

## 1 Introduction

Drifters, devices that naturally float and follow the surface flow lines in a moving water body, are a method to measure important parameters such as surface water velocity distributions, breakthrough

curves and mixing parameters. In drifter research it is important to have as many drifters as possible to optimally measure flow patterns (Tinka et al., 2010). Drifters measure their location, almost exclusively using GPS technology and store it on the device, or transmit it using some network connection. Currently drifters are made for specific research questions and are either expensive to purchase or are build in house, requiring detailed knowledge of electrical engineering and embedded software (Network, 2012; Tinka et al., 2013). In general drifters consist of a location device, often GPS, a logger to store the data and/or a communication unit. Recent developments, most notably the rise of the "maker movement" (Chris, 2012) and its ecosystem based on Arduino, Raspberry Pi and other Open Hardware development boards (Banzi and Shiloh, 2014; Foundation, 2018), have made the individual components of a GPS drifter commercially available as plug and play components for hobbyist and developers. This has led scientists to build GPS loggers geared to their own research need at lower costs. Daniel K. and Peter J. (2012) gave an early overview of the possibilities of the Arduino ecosystem as a low cost alternative for expensive scientific instrumentation. Wickert (2014) developed the Alog, a low power Arduino inspired logger system specifically designed with geoscientific fieldwork in mind.

Relating to using Open Hardware to track GPS positions, Cain and Cross (2018) made a low power GPS logger to track Eastern box turtles. Their device stores the data in the local memory of the Arduino microcontroller, requiring retrieval after the experiment and no real time view on where the turtles are. Pham et al. (2013) and Costanzo (2013) both connected a GPS receiver and a GSM modem to an Arduino to track locations of vehicles. In vehicle tracking usually sufficient power is available through the battery of the car. GPS drifters are often used in near shore applications. Drifters are often designed to store information locally, either by using a complete off the shelf GPS logger such as Sabet and Barani (2011) or by building one from (Open Hardware components such as Lockridge et al. (2016a), Lockridge et al. (2016b) and Johnson et al. (2003). For applications where drifters can not be tracked visually for pick-up, wireless solutions are sometimes used. Austin and Atkinson (2004) added the feature to communicate with the drifters over local radio, to keep track of their location for easy pick-up. Trevathan et al. (2012) custom build a suite of near shore bouys that together act as a Wireless Sensor Network (WSN). Tinka et al. (2013) gives an overview of work on drifters (not focussing on low cost) in their introduction and subsequently present their high end, custom designed Floating Sensor Network. For open ocean tracking Cadena et al. (2018) adds an Irridium satellite modem to their drifter.

For fieldwork on the Ayeyarwady and Chindwin rivers in Myanmar we were in need of a low cost, low power GPS drifter that stores its data locally and at the same time transmits its location in real time for retrieval of the device. Given the length scales involved (tens of kilometers of sampling along the river per day) drifters would float out of range of base stations. However, since we are conducting our fieldwork on a stretch of river with ample GSM coverage, using a satellite modem would be costly overkill. In this article we present, to the best of our knowledge, the first complete

Open Hardware GPS drifter design that adds GSM communication to relay its location in real time. It features an (Open Source) online back-end for both real time tracking and data collection.

All of the projects cited above, while build using Open Hardware low cost components, were either too expensive to purchase for our project or required extensive engineering. In this article we furthermore set out to test the question: "Has the Open Hardware revolution progressed to the point that a low power GPS drifter that wirelessly transmits its position can be made from Open Hardware component by most geoscientists ?". The drifter presented in this paper was deliberately designed from easily connected off the shelf components, allowing for easy construction.

This article will explain how we build such a device, details the software designed and present the raw measurements results from the fieldwork on the Irrawaddy river. The detailed analyses of the data falls beyond the scope of this article, which focuses on the development of the data collection devices, but is presented in the MSc thesis of Bakker (2017). We conclude by reflecting on the primary research question about the "build-ability" of our device by geoscientists in the discussion and conclusions.

## 2 Materials

The drifter design can be separated in two parts. Firstly the drifter itself, ie. the material that makes the drifter float optimally given the local flow conditions of the field site. Secondly the electronics that do the GPS localization and communication. We describe the electronics first and the drifter design afterwards.

### 2.1 Electronic Hardware

The design criteria for our device were to develop a system that could:

– Track its own location using GPS.

– Upload its location to the internet to both save the measurement data and to be able to locate the device in real time during the measurement campaign.

– store the locations locally in case of no connection to the internet.

– operate for at least two days continuously without losing power.

We decided to buy a ready made solution for each design criteria and connect those together. The electronics that matched our design criteria and we choose are:

– a Particle Electron. A development board with build in GSM modem. The electron comes with a data subscription with near global coverage and an online cloud back-end. Users can program the electron with their own code using an online development environment (Particle, 2018) in the C++ dialect that is also used for the Arduino ecosystem (Arduino, 2018). The

Particle Electron is Open Hardware par (2020b) and the Operating system running on the
     device is Open Source software par (2020a).

   – a Particle Asset Tracker Shield, an extension to the Particle Electron, that includes a GPS
     receiver and an accelerometer. The Particle Electron connects to this shield using a set of
     headers on the shield. Communication between the Electron and the GPS receiver is done
over UART Serial communication using the Tx and Rx pins of the Electron.

   – a Sparkfun OpenLog. A board that records any data send to it using UART serial communica-
     tion onto an SD card. While the Particle Electron does have 80 kb of EEPROM memory that
     could be used for storing measurements, adding a removable SD card makes it easy for any
     team member during field work to offload the measurement data.

– an $8 \times 12$ centimeter solar panel.

Table 5 list online locations where the components were purchased and prices at the time of writing.
All datasheets with detailed technical specifications of the components are available as supplemen-
tary material to this article. Connection of the components to each other is straightforward and given
in table 5.

**2.2   Software and online back-end**

The software required to operate the GPS drifters is divided into two parts: the software that runs on
the Particle Electron and the software that runs online. All code used is available on Github and has
been Archived on Zenodo (Hut, 2018). This separation between online and firmware was made to
facilitate re-use by researchers that want to build on one, but not both, of these parts.

Just as with its spiritual predecessor the Arduino, the Particle Electron stores a single program in
its program memory and this program is executed continuously as soon as the Particle Electron is
powered. The Particle Electrons is programmed in the same C++ dialect as is used for the Arduino.
Ready made examples exist that let the Particle Electron and the Asset Tracker Shield upload GPS
locations on a regular bases to the Particle Cloud service. However, these programs keep the cellu-
lar connection open continuously which drains too much power for our application. The code was
heavily modified to optimize for minimal power consumption. Figure 5 shows the architecture of
the code that runs on the Particle Electron. Every minute the device wakes from a low power deep
sleep mode and records the GPS coordinates from the Asset Tracker Shield. This is stored both in
the volatile memory of the device as well as on the SD card in the OpenLog. Once every fifteen
minutes the device will try to connect to the GSM network. If successful, it will upload the 15 latest
GPS measurements to the Particle Cloud. The code is provided on Github and archived on Zenodo
(Hut, 2018).

Figure 5 shows how the data moves from the Particle Electron, via the Particle Cloud service unto
a personal webserver. A webhook running on the Particle Cloud service parses the messages from

the Particle Electron and posts it to the 'receiveData.php' file on the server. This file first stores the raw data in a dataBase, as a backup measure. Than it parses the message into the individual GPS measurements and stores these in the dataBase labelled 'Processed Data'. Users can look at this data in two ways. By opening the file dataDump.php in a browser a csv dump from the dataBase is generated. This allows researchers to do analyses on the measurements after the field campaign.

When realTimeMap.html is opened a single device can be selected from a drop-down box. Once selected a map with the latest measurements from that device is shown. This allows researchers to track the devices in real time. All source code that runs on the personal server, including instructions how to install the Particle webhook, is provided on github and archived on Zenodo (Hut, 2018).

### 2.3   Power consumption

A critical design criteria for our devices is that they should be able to run continuously on the combination of solar panel and battery. This is not an uncommon constrain for equipment in geoscientific applications. The logging of battery level is an additional feature build into the software running on the Particle Electron. We implemented a deep sleep cycle, where the microcontroller unit (MCU) goes into a deep sleep mode which uses less than 200 micro amperes in between taking measure-

ments from the GPS module. In deep sleep, the GPS unit remains active because it can take up to 20 minutes to achieve a fix on the GPS satellites, a process that takes considerable more power than maintaining this fix.

    We based our decisions relating to power consumption on the specified data-sheets of the hardware involved, others such as Par (2016) have verified that the actual power consumption is close enough

to data-sheet specifications for design decisions. Testing in the field showed that the device functions for at least two full days on a 1200 mAh battery. The addition of the chosen solar panel extends this to about 4 days, depending on local solar conditions. For applications where longer periods of measurement are needed, adding batteries to the Particle Electron is straightforward. Given the low power consumption, simple batteries can easily extend the measurement period by several days.

### 2.4   Physical construction

    The Particle Asset Tracker comes delivered with a water proof enclosure. All electrical components except the solar panel fit inside this enclosure. A small hole was drilled in the enclosure to feed the connections from the solar panel to the Particle Electron. The hole, and the connectors on the solar panel, were sealed with two component epoxy to make them waterproof. This was chosen over more

conventional closures such as hot-glue because of its excellent thermal properties: once hardened it has negligible expansion when exposed to temperature differences.

    We designed, based on local available material a wooden plate with Styrofoam and 5 anchors to make sure the drifter followed the flow of the river. In our condition, with low wind conditions this

design worked adequately for our experiment, but of course design of the floaters need to be tailor made based on local conditions and scientific questions.

## 3 Fieldwork: test in Ayeyarwady

Experiments were conducted both upstream of the confluence in the Ayeyarwady as well as at the Chindwin-Ayeyarwady confluence. The Ayeyarwady river system is a very dynamic and naturally, unregulated, river and one of the largest rivers in the world and a vital vein for Myanmar (SOBA, 2018). It not only is one of the main transport corridors, it is also the source of fish, irrigation and drinking water and an important aquatic ecosystem (SOBA, 2018). It is a shallow dynamic river system, typically a depth range of 1-10 m only whereas it has a width of several 100 m. The migrating sandbanks, especially upstream and around the confluence with the Chindwin, are a very pressing problem for river management. Insight in hydraulic behaviour of the river is therefore a continuous demand. First of all, groups of five drifters were release at the same time in a line across the river. The purpose of this experiment was to validate a numerical study on the surface water flow velocity of the river. This analyses falls outside of the scope of this article and is presented in Bakker (2017) in which it is concluded that: "the floater paths around the confluence have been useful for validating the Delft3D model, as the floater speeds and paths observed in the field were compared to those of the Delft3D model." (Bakker, 2017)

The drifters were followed, both online and with a trailing boat. For logistical reasons, when all drifters became too dispersed along the river length (not width), they were taken out of the water and brought together to restart the experiment. If a single drifter started significantly lacking behind with respect to the rest of the group, it was retrieved.

The upstream Ayeyarwady experiment was executed with a single group of five drifters. The confluence experiment was executed with two groups of five drifters upstream of the Chindwin-Ayeyarwady confluence, one in each river. The drifters were tracked along the confluence. At the end of the experiment, drifters were collected and data was read out both from the SD-cards of the OpenLog as well as from the website to which the data had been uploaded.

The raw data collected from the sensors are plotted on a map of the experiment area, see Figure 5. No post-processing has been done on these results: they show the raw GPS locations as recorded by the sensors and downloaded from the online database. The figure has been made by loading these measurements in Google Earth. The different flow paths that the sensors followed can clearly be identified. These data being available online directly allowed the measurement team to identify trackers that had beached, or were stuck in near shore eddies. The full raw dataset is provided as supplementary material to this article. This raw data will be used in follow up research. Figure 5 furthermore shows a closeup of the upstream Ayeyarwady experimental results. In this experiment the floaters were in the water for 16:49 hours and travelled 50km in that time, with a distance of 4.6

km between the fastest and slowest drifter. Mean floater velocity over this stretch was 0.8 meter per second (Bakker, 2017). The location of the drifters clearly shows the different flow paths in the river. Unfortunately, we do spot some gaps in the data. These are caused by lack of cell coverage. In these instances, the Particle Electron took very long in trying to connect, before deciding the switch to offline leading to a loss of data. Based on time between signals that were received, we estimate that around ten percent of data was lost in this manner, although differences between different trackers were significant. Future versions of the software code running on the Particle Electron should switch sooner to offline when cell coverage drops to prevent loss of data. Finally, although clearly labeled as experimental equipment, some sensors were picked out of the water by citizens. This was easily identified in the online visualization (ie. sensors moving upstream) and allowed the research team to chase and recover those sensors. These instances were manually removed from the dataset.

## 4  Discussion

The intention of this research was to test if currently available plug and play sensors and communication solutions are such that most members of the geoscientific community can use them to build their own measurement equipment suited to the specifics of the measurement campaign. Our experience shows that many geoscientifically trained persons should be able to connect the different pieces together and build the basic drifters. However, to have the entire setup operate at the required low power levels requires changing the standard programs that run on the device. The level of understanding of the electronic functioning of the device is not something that in general can be expected from a typical geoscientist, even one with extensive experience in using various types of electronic measurement equipment. Furthermore, developing the online back-end that visualizes the location of the GPS drifters in real time required back up from colleagues (see acknowledgements) with web-development expertise. Besides the design of the drifter, also the construction of the trackers in Myanmar required creativity of using local available materials as well as the expertise of the design to be able to fine-tune and debug the tracker in the field. In our case, the code was changed such that if no cell signal was found, the logger would check again in five, instead of one, minutes, to minimize the data gaps (See above). Also, revitalising a drowned tracker (due to collision with a commercial shipping vessel) needed engineering expertise. These type of experiences learned us that the real-world application of self-designed Open Hardware equipment is not that straightforward as sometimes envisioned. Here we advocate that multi-disciplinary collaboration is still required when using low-cost self designed equipment. Making the design more robust will come against higher expenses.

Being able to trace the low-cost GPS drifters in real time proved invaluable during the fieldwork, for example in the case when curious locals picked up our drifters, as explained above. The real time visualization allowed us to recover equipment.

## 5 Conclusions

We presented a low cost GPS drifter that communicates its data in real time over the cellular network. We presented our design such that fellow geoscientist can (re)build or modify our GPS drifters using this article. Our device was tested in an extensive fieldwork in the Chindwin and Ayeyarwady rivers in Myanmar. During development we tested if such a device can be completely developed by a geoscientist from available plug and play components. Although creating an online logging GPS

drifter turned out to be do-able without electrical engineering background, it required considerable understanding of embedded programming to develop a version that was low power enough to be useful in our fieldwork environment. It would be extremely useful for the geoscientific community if low power versions of the electronics used in this research were available "out of the box". Several companies are building lower power logging equipment for geoscientists Wickert (2014), but these

are usually not yet usable without having to program them first.

The online environment that visualizes the field work data in real time as it comes in was developed by the main author and hosted on his personal website. It would be a great asset to geoscientists doing fieldwork if there was a platform that both stored and visualizes real time data coming in. Part of this functionality is build in the CUAHSI Hydroshare project Tarboton et al. (2014), although this

focuses more on storing data for prosperity than on real time checking of data as it comes in.

The advance of plug and play electronics has opened the opportunity for geoscientists to patch together the functionality they need in a sensor-logger-communication device. We showed that for a GPS drifter with cellular connectivity and local storage this is indeed possible. However, getting plug and play devices to be low power enough for fieldwork condition still requires considerable

electrical engineering knowledge. We hope that this research provides a step in the right direction to help geoscientists use our GPS drifters in particular and plug and play devices in general and encourage the companies producing these devices to bring low power versions to market that work out of the box.

*Acknowledgements.* The authors would like to thank MSc and PhD students from YTU and MMU (Yangon,

Myanmar), MTU (Mandalay, Myanmar) and TUDelft (The Netherlands), for their assistance in the field. Futhermore we like to thank MSc student Rick Hagenaars for helping developing the online back-end, proving the value of having students with a diverse background and skill set in your team / community.

Finally, the authors would like to thank VPDelta for their support, and in particular the Partner voor Water project: "Leapfrogging Delta Management in Myanmar - Showcase smart information solutions in the

Ayeyawady Delta" and NICHE-NMR-250 "Capacity Development on Integrated Water Resources Management in Myanmar"

## Author contribution

Author contributions are listed using the CASRAI Credit system Allen et al. (2014)

– Rolf Hut: Conceptualization, Methodology, Investigation, Resources, Software, Validation, Visualization, Writing – original draft, Writing – review and editing.


– Thanda Thatoe Nwe Win: Methodology, Investigation, Project administration, Writing - original draft (section fieldwork: test in Ayeyarwady).

– Thom Bogaard: Conceptualization, Investigation, Resources, Supervision, Writing – review and editing.

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

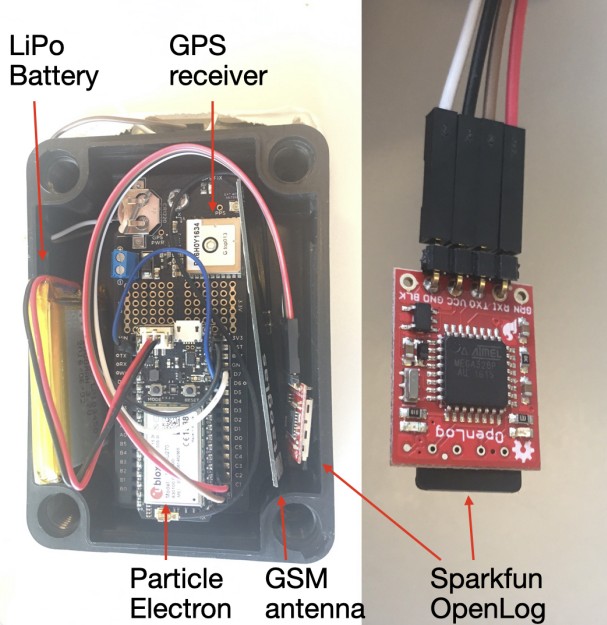

**Figure 1.** Photo of the electronics components. The left photo identifies the Asset Tracker, Electron and Open-Log. The right close ups show the OpenLog in detail.

Sabet, B. S. and Barani, G. A.: Design of Small GPS Drifters for Current Measurements in the Coastal Zone, Ocean & Coastal Management, 54, 158–163, doi:10.1016/j.ocecoaman.2010.10.029, 2011.

SOBA: The Ayeyarwady State of the Basin Assessment (SOBA), https://www.myanmarwaterportal.com/news/663-the-ayeyarwady-state-of-the-basin-assessment-soba.html, 2018.

Tarboton, D. G., Idaszak, R., Horsburgh, J. S., Heard, J., Ames, D., Goodall, J. L., Band, L., Merwade, V., Couch, A., Arrigo, J., et al.: HydroShare: Advancing Collaboration through Hydrologic Data and Model Sharing, 2014.

Tinka, A., Strub, I., Wu, Q., and Bayen, A. M.: Quadratic Programming Based Data Assimilation with Passive Drifting Sensors for Shallow Water Flows, International Journal of Control, 83, 1686–1700, doi:10.1080/00207179.2010.489621, 2010.

Tinka, A., Rafiee, M., and Bayen, A. M.: Floating Sensor Networks for River Studies, IEEE Systems Journal, 7, 36–49, doi:10.1109/JSYST.2012.2204914, 2013.

Trevathan, J., Johnstone, R., Chiffings, T., Atkinson, I., Bergmann, N., Read, W., Theiss, S., Myers, T., and Stevens, T.: SEMAT — The Next Generation of Inexpensive Marine Environmental Monitoring and Measurement Systems, Sensors, 12, 9711–9748, doi:10.3390/s120709711, 2012.

Wickert, A. D.: The ALog: Inexpensive, Open-Source, Automated Data Collection in the Field, The Bulletin of the Ecological Society of America, 95, 166–176, doi:10.1890/0012-9623-95.2.68, 2014.

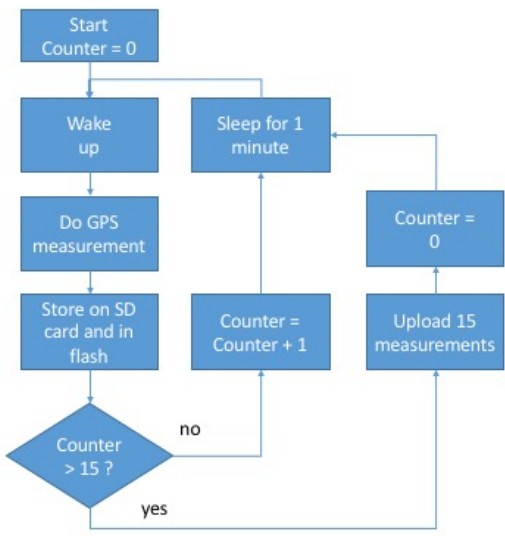

**Figure 2.** a Flowchart of the software that runs on the Particle Electron. The full code is available on Github (Hut, 2018). In situation where limited power is not an issue, this software can be greatly simplified by removing the part where the Electron goes into power saving deep sleep mode. For our case study, however, low power usage was essential to a successful field campaign.

**Table 1.** Price and online location of information on off the shelf electronical components used to build the Low power GPS drifters with local storage and GSM modem.

| Name | Price | Online location |
| --- | --- | --- |
| Particle Asset Tracker | $ 109 | https://www.particle.io/products/hardware/asset-tracker |
| Particle Electron | included with Asset Tracker | https://www.particle.io/products/hardware/electron-cellular-dev-kit |
| SparkFun OpenLog | $ 14.95 | https://www.sparkfun.com/products/13712 |
| Conrad Solar Panel | € 14.95 | https://www.conrad.nl/nl/polykristallijn-zonnepaneel-09-wp-6-v-110454.html |

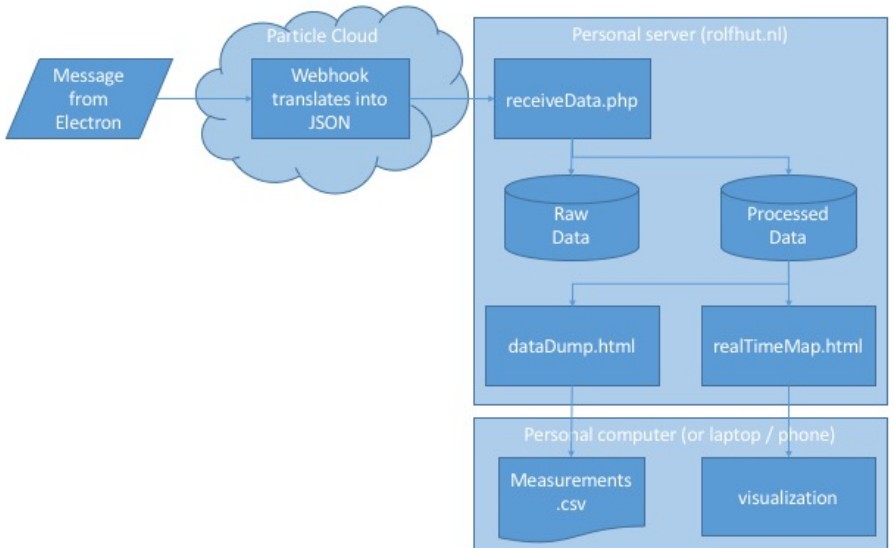

**Figure 3.** The flow of data once it leaves the Particle Electron. The Electron sends the measurements as a comma seperated message to the Particle Cloud. A webhook running on the Particle Cloud parses the raw message into a JSON message that is posted to a php file on the personal server of the main author. This file adds the data to two databases: one to store all raw messages, one that parses the message into the individual measurements. Two html files can be called by users. DataDump.html transforms the database in a comma seperated file for downloading. realTimeMap.html shows the last known locations of the drifters for real time localization during the field campaign. All software is available on github (Hut, 2018)

**Table 2.** Electrical connections between the Particle Asset Tracker and the other components. Note that the connection between GND and OpenLog BLK can be made by connecting the OpenLog pins BLK and GND, saving the need for one wire. The connection between pin D6 and GND is to force the GPS unit on. See figure 5 for an image of the physical setup.

| pin on AssetTracker | connected to |
|---|---|
| power terminal - | solar panel - |
| power terminal + | solar panel + |
| C2 | OpenLog TX0 |
| C3 | OpenLog RX1 |
| GND | OpenLog GND |
| GND | OpenLog BLK |
| 3V3 | OpenLog VCC |
| D6 | Asset Tracker GND |

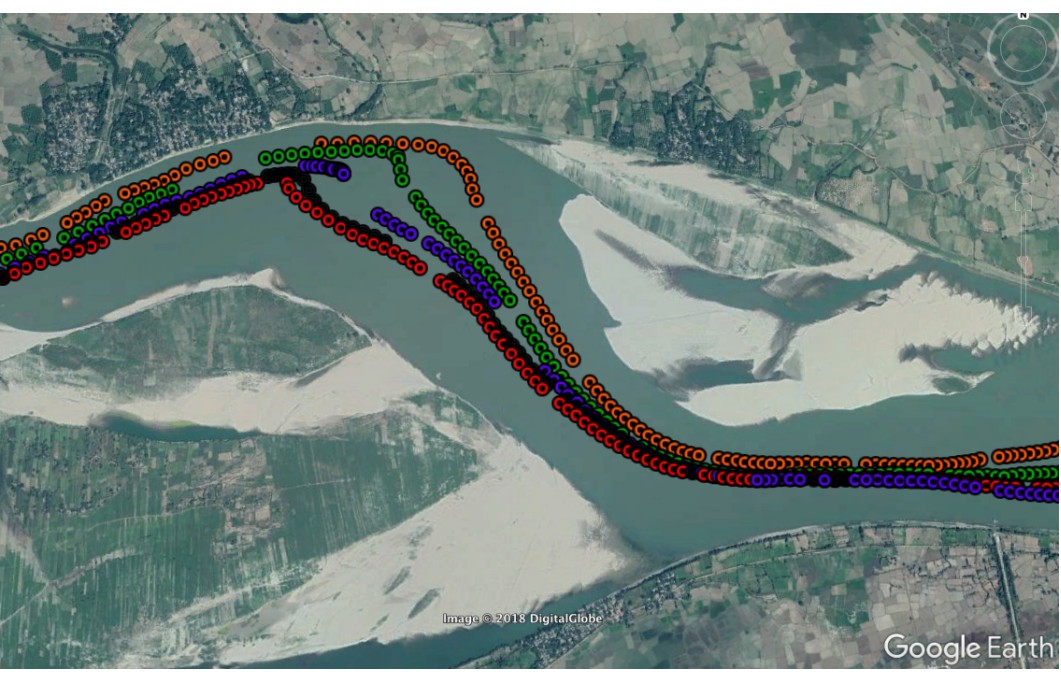

**Figure 4.** The location measurements for the different floaters (different colors) in the Irrawady river near Ngazun. The river flows west to east on this stretch, with downstream being the left side of the figure. ©**Google Earth 2019**

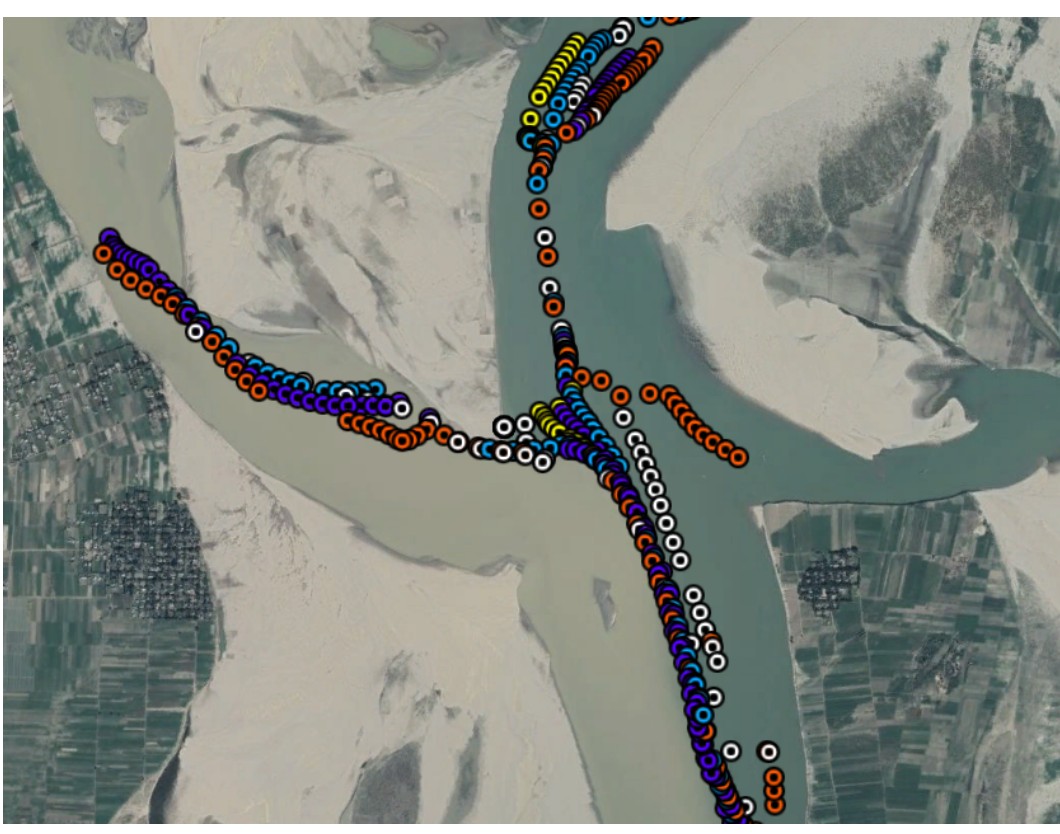

**Figure 5.** The location measurements for the different floaters (different colors) at the confluence of the Chindwin and Irrawaddy rivers. Both rivers stream north to south on this stretch with downstream being the lower part of the figure. **©Google Earth 2019**