# Peer review of "Easy to build Low power GPS drifters with local storage and GSM modem made from off the shelf components"

_Geoscientific Instrumentation, Methods and Data Systems, 2019_

## Referee Comment (RC1) · Anonymous Referee #1 · 29 Nov 2019

General comments:

This manuscript describes the building of a low-cost GPS drifter, and provides the instructions to assemble the device from off-the-shelf components, and to program the microprocessor. The manuscript also provides some broader discussion of the concept of "Open Hardware", and its relevance for scientific practice.

Although I appreciate the authors' clear enthusiasm for low-cost modular and (partly) open source technology that promotes "tinkering", I have several reservations with the current manuscript.

First of all, the scientific content is extremely thin. From a technical perspective, the presented solution is very simple. Literally hundreds of examples of similar (and often

far more sophisticated) designs can be found online, especially within the user communities of the presented technologies. This does not mean that documented another one has no value, but in this case the presented design fails to push sufficiently the technical or scientific boundaries to warrant publication in a peer-reviewed journal (see below for some specific comments on this aspect).

I understand the simplicity of the design is part of the focus of the article, as a way to show how these technologies are becoming more accessible for geoscientists. In itself, I think that this is a valid focus. Technologies such as Arduino have indeed made hardware (and related software) more accessible via a combination of open source licenses, online availability, and modular hardware components. But also here, the manuscript does not endeavor beyond the most well known and obvious.

On the contrary, I find some aspects of this discussion problematic. One is of an ideological nature: as far as I know, the particle.io is not fully open source. Contrary to Arduino, I do not think that the firmware nor the compiler chain are open source (though I would be happy to be proven wrong). They also rely on a commercial telemetry platform, which seems to go against the open source ideology. So it is a bit of an odd choice for a project that advocates open source.

Another aspect is more theoretical: the manuscript seems to divide (geo)scientists in a rather black-and-white fashion into 2 categories, i.e. those with electrical and software engineering skills, and those without. I think that reality is far more nuanced and very different processes are at play here.

One is the open source philosophy, which has indeed promoted accessibility and affordability. Another aspect is simply technological advances - the open source movement does not have an exclusive access to low-cost technology; there is nothing that intrinsically prevents commercial entities to make products of similar levels of cost (and reliability). The fact that this may not happen is probably more an economic question (e.g., lack of sizeable market) than a technological one.
Lastly, the aspect of easy-of-use is much more complex than the authors suggest. Surely the current design is easier to build now than 20 years ago - but it still requires a level of non-trivial expertise that may or may not be available in a research team. The choice for a specific technology will depend on the specific project needs, as well as the available skills, time, and resources available. The availability of open hardware is definitely a useful addition to widen the technological options in this context, but the whole process is probably more complex than the "cheaper-is-better" optimism that this manuscript exudes.

To conclude, the manuscript scratches a surface beyond which there is a lot of interesting science to explore, both in terms of technology and the broader application context. But unfortunately the current manuscript stays very close to that surface.

Specific comments:

- Power consumption: the abstract mentions that making such devices operate at low power levels requires detailed electric expertise. However, this is not elaborated in the manuscript. The manuscript itself is very brief on power consumption, with as main message that the sleep current is around 200uA. That is in itself not very spectacular - many microCPUs consume only a few uA in sleep modes. I assume that in this case the main power consumption is the GPS (although it would be nice to see some specific results). I also wonder whether the detailed electric expertise refers to putting the CPU in sleep mode (which seems pretty straightforward with a single command in the code) or whether it refers to bringing the consumption down below the current 200uA. If the latter, then it would be good to elaborate further. If this is what a GPS needs to maintain a fix on satellites, then it may be very hard to reduce this, even with detailed electric expertise.

- why do you use an SD card? An EEPROM or flash memory chip should provide sufficient storage at a lower cost and lower power consumption.

- is the solar panel really relevant? If it only extends the battery life of the device with

2 days, then adding a second battery seems to be a more logical option. Additionally, a battery life of 2 days seems short if the standby power consumption is only 200uA. Also here, a more detailed assessment of power consumption would be useful.

---

## Referee Comment (RC2) · Anonymous Referee #2 · 5 Jan 2020

The authors present instructions for constructing a low-power GPS drifter with both local data storage and GSM-based data submission to a server. This is an interesting research topic as it can help provide more detailed description of river hydrodynamics over long river reaches, using Lagrangian approach. I believe that this paper is within the scope of the journal.

===

General comments:

The authors provide an enthusiastic description of their approach, with a sufficiently detailed recipe for recreating their drifter design. However, two things have bothered me greatly while reading the manuscript:

1. Construction of a low-cost drifter with of-the-shelf components is not a novel idea. It has been done a good number of times before, especially in marine research (Johnson at al., 2003; Austin and Atkinson, 2004; Sabet and Barani, 2011). Additionally, some articles already offer VERY similar design ideas (Lockrigde et al., 2016; Cadena, Vera and Moreira, 2018). It is very concerning that such similar concepts were not acknowledged in this manuscript. Certain existing approaches describe similarly priced drifters, while also providing tracking of additional parameters – temperature, conductivity, etc. This indicates insufficient literature review prior to the research itself.

2. The amount of scientific novelty in the manuscript appears to be relatively low, considering the state-of-the-art. The manuscript only offers somewhat different technical solution for the drifter design and a server front/backend, which, by itself, does not merit an original research article. A good article should present the current knowledge on the subject at hand, as well as its limitations, and then clearly state how it aims to expand it and why. I feel that this was not done adequately in this version of the manuscript. One way to solve this issue would be to review the Introduction section to include additional relevant research, to discuss the strengths and weaknesses of different approaches, and then aim to overcome limitations or to improve specific aspects of said research. In order to deliver sufficient scientific novelty, instead of a single design (Particle), authors could perform a detailed comparison of several different designs: Particle-based, Arduino and Arduino-compatible boards, and some commercial/proprietary GPS trackers. This comparison could cover different performance aspects such as accuracy, reliability, battery life, etc. Such review would enable the reader to understand pros and cons of different solutions and choose the one that suits him best This way, the manuscript itself would boast considerably more scientific "weight". In the current form the manuscript would perhaps be a better fit for a "Technical note" article which is available in some journals. This is also supported by the relatively short length of the manuscript.

===
Specific comments:

1. Abstract does not appear to contain all relevant information about the manuscript. A well-designed abstract should be able to present the following: (1) the description of the problem at hand, (2) an explanation on the motivation for solving that specific problem, (3) present an original method for solving that problem, (4) present some of the results and (5) implications of the research. Points (1), (2) and (5) are relatively evident in the current version, but I feel that points (3) and (4) are insufficiently covered. It would be good to explain how the authors' approach is different/better than previous methods. What will this approach enable in the future? Is there a technical improvement other than the use of open hardware/software? The price itself should not be a sole focus, as some existing papers describe similarly priced drifters.

2. Why is a 2-day battery a design criterion? Is there a specific reason for just 2 days? Why not more? How many batteries have been used for powering the board? For most boards, at least 3.3 V are needed for power supply, which requires a minimum of 2x1.2 V batteries. Also, adding a solar panel seems more complicated and unreliable than using a higher capacity battery, since the described battery capacity of 1200 mAh is fairly low. Modern day AA sized NiMH rechargeable batteries have capacities of up to 3 Ah, which would (in theory) increase the battery life to approx. 5 days, thus eliminating the need for a solar panel.

3. A more detailed investigation on power consumption during operation and in deep-sleep mode could be performed and presented. This would allow an interested reader not only to plan the battery lifespan in his own experiment, but also to understand how to potentially improve the power efficiency. Is it possible to measure the consumption and document it in the following version of the manuscript?

4. Line 145 states that the use of five drifters enabled the authors to "quantify the cross-sectional variation in surface water flow velocity". This is a very problematic statement from a hydraulic standpoint as five tracers across a river section is hardly enough for

an adequate estimation of cross-sectional surface velocity distribution. Additionally, as evident in Figures 4 and 5, surface tracers (once released) have quickly converged together and cover less than half of the cross-section. Some researchers have tried verifying the obtained velocity results, for example using ADCP.

===

Technical corrections:

1. There are too many small but evident writing mistakes and inconsistencies. These kinds of mistakes severely decrease the quality of the manuscript as they show negligence in the writing process. A good research should also be carefully and clearly presented to fellow scientists. Some (but likely not all) of the mistakes in question are listed below:

a) Lines 3, 12: "tinkering scientist" was sometimes written with quotation symbols, and sometimes without – this should be consistent. Also, what is the definition of the "tinkering scientist"? The term is perhaps too colloquial to be used in a research article.

b) Line 8: Quotation opened but never closed.

c) Line 9: "wireless" should be "wirelessly".

d) Line 15: "mircro" should be "micro".

e) Line 48: Repetition (from the abstract) – this should be avoided.

f) Lines 56, 114, 125, 135, 169: Headings should start with an uppercase letter.

g) Line 125: Typo in the heading.

h) Line 251: Wrong citation formatting.

2. There are many colloquial terms and ambiguities in the manuscript:

a) Line 118: "MCU" is an abbreviation which was never explained prior to this appearance. I assume it means "microcontroller unit", but all abbreviations (except for famous
ones, such as DNA, GPS, . . .) should be explained when they first appear in the text.

b) Line 132: ". . .to give it drag. . ." is a bit too colloquial.

c) Line 133: ". . .this design worked fine. . ." – adjective "fine" should have no place in a scientific article. Also, "fine" relative to which criteria?

d) Lines 147-8: "fished out" should be "retrieved", or something similar.

e) Line 177: "typical geoscientist" – what is a typical (geo)scientist? This is mentioned in several forms throughout the manuscript, and I feel that this is an unsubstantiated generalization; scientist performing data acquisition is likely to have some degree of knowledge on relevant hardware and software. Given the sheer size of available documentation, online tutorials and dedicated tech-blogs, it is highly unlikely that a team of modern-day scientists would have significant problems constructing a modular, microcontroller-based piece of equipment (e.g. Arduino or similar).

f) Lines 190-4: Repeats a part of section 3, lines 165-7.

g) Line 194: Term "GeoChasing" is not explained. You should provide all relevant information or point to other sources which provide this information in sufficient level of detail. Nothing should be left unexplained ("hanging") in the text.

3. Figures should be of better quality. More specifically:

a) Figures 2 and 3 are of very low resolution – hard to read on-screen, and unintelligible once printed. Font size in figures should generally correspond to the font size in the main text.

b) Figure 1 would benefit from a clear identification of different components.

c) Figures 4 and 5 should have some indication of flow direction. I assume that in Figure 5 the bottom part of the image is the upstream region, but this should be clearly indicated.

4. Why is Line 275 after Figures 1 and 2 and not with the other references?

===

References:

Johnson, D., Stocker, R., Head, R., Imberger, J. and Pattiaratchi, C., 2003. A compact, low-cost GPS drifter for use in the oceanic nearshore zone, lakes, and estuaries. Journal of atmospheric and oceanic technology, 20(12), pp.1880-1884.

Austin, J. and Atkinson, S., 2004. The design and testing of small, low-cost GPS-tracked surface drifters. Estuaries, 27(6), pp.1026-1029.

Sabet, B.S. and Barani, G.A., 2011. Design of small GPS drifters for current measurements in the coastal zone. Ocean & coastal management, 54(2), pp.158-163.

Lockridge, G., Dzwonkowski, B., Nelson, R. and Powers, S., 2016. Development of a low-cost Arduino-based sonde for coastal applications. Sensors, 16(4), p.528.

Cadena, A., Vera, S. and Moreira, M., 2018, April. A low-cost Lagrangian drifter based on open-source hardware and software platform. In 2018 4th International Conference on Control, Automation and Robotics (ICCAR) (pp. 218-221). IEEE.

---

## Author Comment (AC1) · 29 Jun 2020

**Reply to reviewers**

We thank the reviewers for their constructive feedback on our manuscript. With the help of the reviewers comments, we have improved our manuscript.

From the reviewers comments it is clear that we failed to convince what we believe to be the addition to the state of the art that we want to introduce to the literature with our contribution. The reviewers rightly comment that different drifter design, also ones based on all open hardware components, including once based on Arduino, or Arduino-like, development boards, have been published before. All of these include some form of recording the GPS position of the drifter. However, none of the these drifter solutions incorporates a GSM solution that is easy to install and allows the drifters to be picked up based on their online location. Some solutions available in the literature (mainly for near coast applications) include radio based communication to a local station, but these have limited range and are not suited for the particular use case we are looking at where drifters travel multiple (sometimes tens) of kilometers along a river stretch before being recovered.

We have extensively rewritten the introduction of our manuscript to highlight the existing literature on drifters and to position our contribution among those.

Furthermore, we have added and changed the manuscript based on the specific comments of the reviewers. In the rest of this document, we will explain how we incorporated the comments of the reviewers. Any texts in *italics* are the original comments of the reviewers.

**Reviewer nr 1.**

**General comments:**

This manuscript describes the building of a low-cost GPS drifter, and provides the instructions to assemble the device from off-the-shelf components, and to program the microprocessor. The manuscript also provides some broader discussion of the concept of "Open Hardware", and its relevance for scientific practice.

Although I appreciate the authors' clear enthusiasm for low-cost modular and (partly) open source technology that promotes "tinkering", I have several reservations with the current manuscript.

First of all, the scientific content is extremely thin. From a technical perspective, the presented solution is very simple. Literally hundreds of examples of similar (and often far more sophisticated) designs can be found online, especially within the user communities of the presented technologies. This does not mean that documented another one has no value, but in this case the presented design fails to push sufficiently the technical or scientific boundaries to warrant publication in a peer-reviewed journal (see below for some specific comments on this aspect).

I understand the simplicity of the design is part of the focus of the article, as a way to show how these technologies are becoming more accessible for geoscientists. In itself, I think that this is a valid focus. Technologies such as Arduino have indeed made hardware (and related software) more accessible via a combination of open source licenses, online availability, and modular hardware components. But also here, the manuscript does not endeavor beyond the most well known and obvious. On the contrary, I find some aspects of this discussion problematic. One is of an ideological nature: as far as I know, the particle.io is not fully open source. Contrary to Arduino, I do not think that the firmware nor the compiler chain are open source (though I would be happy to be proven wrong). They also rely on a commercial telemetry platform, which seems to go against the open source ideology. So it is a bit of an odd choice for a project that advocates open source.

We agree with the reviewer that our article focusses on introducing (yet) another drifter to the academic literature. As explained above: what we aim add to the state of the art is that our drifter uses an easy to use GSM platform, making it ideally suited for along river experiments by groups that do not have the technical know-how to set up their own GSM modem service (we will address the technology efficacy comment below). We have stressed this on numerous places throughout the new manuscript, for example lines 64-65, in the abstract and the title.

With respect to the Particle platform being Open Source, to the best of our knowledge, the parts of the platform that we use, are open source: the Electron is Open Hardware (https://github.com/particle-iot/electron, CC-BY-SA-4.0), the device OS is in (very) active development by both Particle itself and the community (https://github.com/particle-iot/device-os) including a guide on how to install the compiler chain locally (https://github.com/particle-iot/device-os/blob/develop/docs/dependencies.md). We want to stress that "full open source" is somewhat of a "turtles all the way down" issue (the lithographic masks to make the atmega328p used on the Arduino Uno is not publicly available for example). To our knowledge, the hardware and software used in our study fall within the broadly accepted definition of "Open Source Hardware".

We have provided references in the manuscript to these repositories and highlighted the Open Source nature of the products used.

Another aspect is more theoretical: the manuscript seems to divide (geo)scientists in a rather black-and-white fashion into 2 categories, i.e. those with electrical and software engineering skills, and those without. I think that reality is far more nuanced and very different processes are at play here. One is the open source philosophy, which has indeed promoted accessibility and affordability. Another aspect is simply technological advances - the open source movement does not have an exclusive access to low-cost technology; there is nothing that intrinsically prevents commercial entities to make products of similar levels of cost (and reliability). The fact that this may not happen is probably more an economic question (e.g., lack of sizeable market) than a technological one. Lastly, the aspect of easy-of-use is much more complex than the authors suggest. Surely the current design is easier to build now than 20 years ago - but it still requires a level of non-trivial expertise that may or may not be available in a research team. The choice for a specific technology will depend on the specific project needs, as well as the available skills, time, and resources available. The availability of open hardware is definitely a useful addition to widen the technological options in this context, but the whole process is probably more complex than the "cheaper-is-better" optimism that this manuscript exudes.

We agree with the reviewer that the artificial divide between "those with electrical and software engineering skills" and those without needs more nuance. The skill level in electrical and software engineering in geoscientists does indeed cover a broad spectrum. What is "adequate" sensing equipment will differ based on skill levels present within a research team and on available budget. We like to stress that for research teams with

- a minimum, but not a deep, knowledge of electrical engineering
- programming skills, but not deep software design expertise
- high level understanding of networking, but no detailed knowledge of network layers / IP-addresses of DNS servers

- moderate (NSF levels) but not Bill Gates endowment levels of funding the technology presented in our manuscript adds a piece of useful tooling. We have reworded the parts of the manuscript where the difference between geoscientists with and without electrical and software engineering skills was presented too stark and have focused on what the solution offers, rather than what people might be missing in skills.

To conclude, the manuscript scratches a surface beyond which there is a lot of interesting science to explore, both in terms of technology and the broader application context. But unfortunately, the current manuscript stays very close to that surface.

We hope that with the changes in the manuscript we have convinced the reviewer the added value of our contribution to the scientific literature.

**Specific comments:**

Power consumption: the abstract mentions that making such devices operate at low power levels requires detailed electric expertise. However, this is not elaborated in the manuscript. The manuscript itself is very brief on power consumption, with as main message that the sleep current is around 200uA. That is in itself not very spectacular - many microCPUs consume only a few uA in sleep modes. I assume that in this case the main power consumption is the GPS (although it would be nice to see some specific results). I also wonder whether the detailed electric expertise refers to putting the CPU in sleep mode (which seems pretty straightforward with a single command in the code) or whether it refers to bringing the consumption down below the current 200uA. If the latter, then it would be good to elaborate further. If this is what a GPS needs to maintain a fix on satellites, then it may be very hard to reduce this, even with detailed electric expertise.

The main power consumption is indeed the GPS (based on datasheet information). Putting the CPU in sleep looks like one line of code, but introduces issues that many geoscientists venturing into the realm of embedded devices might not recognize, most notably the requirement to use retained variables, since deep sleep mode resets the variable memory. We believe that the reviewer overestimates the level of electrical engineering present in many geoscientific groups. The authors, who have been working on sensing environmental variables using open source hardware for a few years, were stumped by this for quite some time.

Following reviewer #2's comment we realised that studying the power consumption of the device in greater detail would certainly be an interesting addition. We have based our power estimations on datasheet specification of the different components, something that

is available to anyone. Measuring the power consumption during deep sleep and especially connecting to the GSM service would require more sensitive equipment and is beyond the scope of this work where we report on the design, from the datasheet's specs up, of our drifter design. We do, however, want to provide the readership with as much info to base their choice of drifter design on and have added references to online tests done by others on the power consumption of the Partice Electron during deep sleep and when connecting to the GSM network.

why do you use an SD card? An EEPROM or flash memory chip should provide sufficient storage at a lower cost and lower power consumption.

The use of an SD card is from the point of view of easy operation in the field. Any of our students can open the enclosure and swap out an SD card. Had we stored the raw data in the 80 kb available EEPROM memory, we would have needed to write both additional software on the device to allow reading out this data in the field as well as software on our laptops to read out this EEPROM and save it locally. By using an SD card we made this process easier, a guiding principle throughout our project. To alert readers to the option of using the EEPROM (and saving money on the SD card and OpenLog), we added the following line to the manuscript:

While the Particle Electron does have 80 kb of EEPROM memory that could be used for storing measurements, adding a removable SD card makes it easy for any team member during field work to offload the measurement data. [line 95]

Is the solar panel really relevant? If it only extends the battery life of the device with C3 2 days, then adding a second battery seems to be a more logical option. Additionally, a battery life of 2 days seems short if the standby power consumption is only 200uA. Also here, a more detailed assessment of power consumption would be useful.

In lab tests, the battery life was considerably longer than two days. However, in the field this dropped to two days. Looking back, we believe this is because when the device has spotty GSM reception, it takes longer to establish a connection, taking more power and when no connection is available, the device tries to connect more often, also consuming more power. The 2-day battery requirement stems from the specifics of our field campaign: we wanted to be able to let the drifters drift down the river overnight and pick them up the next morning. The Particle Electron is shipped with a 1200 mAh Li-ion battery and we wanted to keep the additional elements to a minimum. The reviewer is correct that adding a few AA batteries greatly improves the operational use and may be preferable for people in other applications. We have added lines explaining this to the methods and material section.

For applications where longer periods of measurement are needed, adding batteries to the Particle Electron is straightforward. Given the low power

consumption, simple batteries can easily extend the measurement period by several days. [line 145]

**Reviewer nr. 2.**

The authors present instructions for constructing a low-power GPS drifter with both local data storage and GSM-based data submission to a server. This is an interesting research topic as it can help provide more detailed description of river hydrodynamics over long river reaches, using Lagrangian approach. I believe that this paper is within the scope of the journal.

**General comments:**

The authors provide an enthusiastic description of their approach, with a sufficiently detailed recipe for recreating their drifter design. However, two things have bothered me greatly while reading the manuscript:

Construction of a low-cost drifter with of-the-shelf components is not a novel idea. Ithas been done a good number of times before, especially in marine research (Johnson at al., 2003; Austin and Atkinson, 2004; Sabet and Barani, 2011). Additionally, some articles already offer VERY similar design ideas (Lockrigde et al., 2016; Cadena, Vera and Moreira, 2018). It is very concerning that such similar concepts were not acknowledged in this manuscript. Certain existing approaches describe similarly priced drifters, while also providing tracking of additional parameters – temperature, conductivity, etc. This indicates insufficient literature review prior to the research itself.

As per the comment at the start of this document, we regret that we failed to show what we believe to be the addition to the state of the art the we are making with our research. The reviewer is right that many others have constructed GPS drifters and we have added those to the literature review in the introduction. Paragraph on others work (lines 35-50) has been extended. What we add to this body of knowledge is a drifter design that adds connectivity over GSM, making it especially suited for studies in long river stretches, without asking the geoscientist who want to use this drifter design to learn everything related to GSM connectivity. To make this clear, the paragraph detailing our addition to the literature now reads:

For fieldwork on the Ayeyarwady and Chindwin rivers in Myanmar we were in need of a lowcost, low power GPS drifter that stores its data locally and at the same time transmits its location inreal time for retrieval of the device. Given the length scales involved (tens of kilometres of sampling along the river per day) drifters would float out of range of base stations. However, since we are conducting our fieldwork on a stretch of river with ample GSM coverage, using a satellite modem would be costly overkill. In this article we present, to the best of our knowledge, the first complete Open Hardware GPS drifter design that adds GSM communication to relay its location in real time. It features an (Open Source) online back-end for both real time tracking and data collection. [lines 51-58]

The amount of scientific novelty in the manuscript appears to be relatively low, considering the state-of-the-art. The manuscript only offers somewhat different technical solution for the drifter design and a server front/backend, which, by itself, does not merit an original research article. A good article should present the current knowledge on the subject at hand, as well as its limitations, and then clearly state how it aims to expand it and why. I feel that this was not done adequately in this version of the manuscript. One way to solve this issue would be to review the Introduction section to include additional relevant research, to discuss the strengths and weaknesses of different approaches, and then aim to overcome limitations or to improve specific aspects of said research. In order to deliver sufficient scientific novelty, instead of a single design (Particle), authors could perform a detailed comparison of several different designs: Particle-based, Arduino and Arduino-compatible boards, and some commercial/proprietary GPS trackers. This comparison could cover different performance aspects such as accuracy, reliability, battery life, etc. Such review would enable the reader to understand pros and cons of different solutions and choose the one that suits him best This way, the manuscript itself would boast considerably more scientific "weight". In the current form the manuscript would perhaps be a better fit for a "Technical note" article which is available in some journals. This is also supported by the relatively short length of the manuscript.

We agree with the reviewer that our manuscript is very close to a "Technical Note", in fact, had GI had a "technical note" as option, we would have used that. Unfortunately, there are no journals in the geosciences that offer such technical note (to our knowledge) with a wide enough readership to still reach a broad geoscientific audience. This is why we choose to submit to GI. With our article we want to give the readership of GI a thoughtful overview of our device, and at the same time place it among the state of the art. We have added significantly to the introduction to indicate where our device adds to the state of the art compared to existing literature.

**Specific comments:**

Abstract does not appear to contain all relevant information about the manuscript. A welldesigned abstract should be able to present the following: (1) the description of the problem at hand, (2) an explanation on the motivation for solving that specific problem, (3) present an original method for solving that problem, (4) present some of the results and (5) implications of the research. Points (1), (2) and (5) are relatively evident in the current version, but I feel that points (3) and (4) are insufficiently covered. It would be good to explain how the authors' approach is different/better than previous methods. What will this approach enable in the future? Is there a technical improvement other than the use of open hardware/software? The price itself should not be a sole focus, as some existing papers describe similarly priced drifters.

See the comment at the start of this document: we regret that we failed to communicate how we add to the existing state of the art. The focus of our manuscript is on the fact that, to our knowledge for the first time, easy to use GSM connectivity has been added to GPS drifters. We have added this to the abstract and on numerous places in the manuscript Why is a 2-day battery a design criterion? Is there a specific reason for just 2 days? Why not more? How many batteries have been used for powering the board? For most boards, at least 3.3 V are needed for power supply, which requires a minimum of 2x1.2 V batteries. Also, adding a solar panel seems more complicated and unreliable than using a higher capacity battery, since the described battery capacity of 1200 mAh is fairly low. Modern day AA sized NiMH rechargeable batteries have capacities of up to 3 Ah, which would (in theory) increase the battery life to approx. 5 days, thus eliminating the need for a solar panel.

The 2-day battery requirement stems from the specifics of our field campaign: we wanted to be able to let the drifters drift down the river overnight and pick them up the next morning. The Particle Electron is shipped with a 1200 mAh Li-ion battery and we wanted to keep the additional elements to a minimum. The reviewer is correct that adding a few AA batteries greatly improves the operational use and may be preferable for people in other applications. We have added lines explaining this to the methods and material section.

A more detailed investigation on power consumption during operation and in deepsleep mode could be performed and presented. This would allow an interested reader not only to plan the battery lifespan in his own experiment, but also to understand how to potentially improve the power efficiency. Is it possible to measure the consumption and document it in the following version of the manuscript?

Studying the power consumption of the device in greater detail would certainly be an interesting addition. We have based our power estimations on datasheet specification of the different components, something that is available to anyone. Measuring the power consumption during deep sleep and especially connecting to the GSM service would require more sensitive equipment and is beyond the scope of this work where we report on the design, from the datasheet's specs up, of our drifter design. We do, however, want to provide the readership with as much info to base their choice of drifter design on and have added references to online tests done by others on the power consumption of the Partice Electron during deep sleep and when connecting to the GSM network.

Line 145 states that the use of five drifters enabled the authors to "quantify the crosssectional variation in surface water flow velocity". This is a very problematic statement from a hydraulic standpoint as five tracers across a river section is hardly enough for an adequate estimation of cross-sectional surface velocity distribution. Additionally, as evident in Figures 4 and 5, surface tracers (once released) have quickly converged together and cover less than half of the cross-section. Some researchers have tried verifying the obtained velocity results, for example using ADCP.

The reviewer is right that on its own, 5 drifters would be too few measurements to quantify cross-sectional variation. We used the drifter experiment to verify predictions by a numerical model of the same river. These model results fall outside of the scope of this manuscript (and are presented in the MSC thesis of Bakker, referenced in the article).

Technical corrections:

1. There are too many small but evident writing mistakes and inconsistencies. These kinds of mistakes severely decrease the quality of the manuscript as they show negligence in the writing process. A good research should also be carefully and clearly presented to fellow scientists. Some (but likely not all) of the mistakes in question are listed below:

a) Lines 3, 12: "tinkering scientist" was sometimes written with quotation symbols, and sometimes without – this should be consistent. Also, what is the definition of the "tinkering scientist"? The term is perhaps too colloquial to be used in a research article.

This is made consistent. As a tinkering scientist, I take pride in that title and would personally not consider it either colloquial or lessening of the work that I do.

b) Line 8: Quotation opened but never closed.

**Changed**

c) Line 9: "wireless" should be "wirelessly".

**Changed**

d) Line 15: "mircro" should be "micro".

**Changed**

e) Line 48: Repetition (from the abstract) – this should be avoided.

**Slightly reworded**

f) Lines 56, 114, 125, 135, 169: Headings should start with an uppercase letter.

**Changed**

g) Line 125: Typo in the heading.

**Changed**

h) Line 251: Wrong citation formatting.

We will coordinate correct formatting with the editorial office for the final version of the manuscript.

2. There are many colloquial terms and ambiguities in the manuscript: a) Line 118: "MCU" is an abbreviation which was never explained prior to this appearance. I assume it means "microcontroller unit", but all abbreviations (except for famous ones, such as DNA, GPS, ...) should be explained when they first appear in the text.

**Changed**

b) Line 132: "...to give it drag..." is a bit too colloquial.

**Changed**

c) Line 133: "...this design worked fine..." – adjective "fine" should have no place in a scientific article. Also, "fine" relative to which criteria?

Changed to "adequately for our experiment"

d) Lines 147-8: "fished out" should be "retrieved", or something similar.

**Changed**

e) Line 177: "typical geoscientist" – what is a typical (geo)scientist? This is mentioned in several forms throughout the manuscript, and I feel that this is an unsubstantiated generalization; scientist performing data acquisition is likely to have some degree of knowledge on relevant hardware and software. Given the sheer size of available documentation, online tutorials and dedicated tech-blogs, it is highly unlikely that a team of modern-day scientists would have significant problems constructing a modular, microcontroller-based piece of equipment (e.g. Arduino or similar).

See the comment to reviewer #1 above

f) Lines 190-4: Repeats a part of section 3, lines 165-7.

The second instance is made much shorter and refers back to the first instance.

g) Line 194: Term "GeoChasing" is not explained. You should provide all relevant information or point to other sources which provide this information in sufficient level of detail. Nothing should be left unexplained ("hanging") in the text.

We apologize for the assumption that GeoChasing was a commonly understood pastime and have removed the sentence.

3. Figures should be of better quality. More specifically:

a) Figures 2 and 3 are of very low resolution – hard to read on-screen, and unintelligibleonce printed. Font size in figures should generally correspond to the font size in the main text.

We upgraded the quality of the figures. Since font size will change between review and final versions of the manuscript, we will coordinate figure quality with the GI editorial team.

b) Figure 1 would benefit from a clear identification of different components.

We added labels to the figure to identify the components

c) Figures 4 and 5 should have some indication of flow direction. I assume that in Figure 5 the bottom part of the image is the upstream region, but this should be clearly indicated.

We added flow direction in the caption of the figures

4. Why is Line 275 after Figures 1 and 2 and not with the other references?

This is a LaTeX issue that we will sort out with the editorial staff of GI in post production.

---

## Author Response (AR2)

Reply to reviewer and editor in final round of review of:

**Easy to build Low power GPS drifters with local storage and GSM modem made from off the shelf components.**

Dear editor and reviewer,

We thank you for this second round of review of our manuscript and are happy to read that the reviewers states: "the more I read through the manuscript, the more I appreciate certain aspects of it."

As asked for by the editor, we have implemented the technical corrections. Furthermore, we address point #6 of the reviewer by extending the results and discussion sections of our manuscript. Although we want to keep the main focus of the manuscript on the (development of) the device and not on the science done with it, on the reviewers request we added additional information from Bakker (2017) on the distances the drifters traveled, the amount of signal loss and the post processing procedure.

During the revision process, we see the manuscript has been improved considerably, due to the constructive feedback by both reviewers, for which we are grateful.

[revised manuscript text omitted]